# Investigation of Graphene Derivatives on Electrical Properties of Alkali Activated Slag Composites

**DOI:** 10.3390/ma14164374

**Published:** 2021-08-04

**Authors:** Wu-Jian Long, Xuan-Han Zhang, Bi-Qin Dong, Yuan Fang, Tao-Hua Ye, Jing Xie

**Affiliations:** Shenzhen Durability Center for Civil and Transportation Engineering, Guangdong Provincial Key Laboratory of Durability for Marine Civil Engineering, College of Civil and Transportation Engineering, Shenzhen University, Shenzhen 518060, China; longwj@szu.edu.cn (W.-J.L.); zhangxuanhan2019@email.szu.edu.cn (X.-H.Z.); incise@szu.edu.cn (B.-Q.D.); yuanfang@szu.edu.cn (Y.F.); yetaohua2017@email.szu.edu.cn (T.-H.Y.)

**Keywords:** reduced graphene oxide, graphene oxide, alkali activated slag composite, rheology, electrical properties

## Abstract

Reduced graphene oxide (rGO) has been widely used to modify the mechanical performance of alkali activated slag composites (AASC); however, the mechanism is still unclear and the electrical properties of rGO reinforced AASC are unknown. Here, the rheological, mechanical, and electrical properties of the AASC containing rGO nanosheets (0, 0.1, 0.2, and 0.3% wt.) are investigated. Results showed that rGO nanosheets addition can significantly improve the yield stress, plastic viscosity, thixotropy, and compressive strength of the AASC. The addition of 0.3% wt. rGO nanosheets increased the stress, viscosity, thixotropy, and strength by 186.77 times, 3.68 times, 15.15 times, and 21.02%, respectively. As for electrical properties, the impedance of the AASC increased when the rGO content was less than 0.2% wt. but decreased with the increasing dosage. In contrast, the dielectric constant and electrical conductivity of the AASC containing rGO nanosheets decreased and then increased, which can be attributed to the abundant interlayer water and the increasing structural defects as the storage sites for charge carriers, respectively. In addition, the effect of graphene oxide (GO) on the AASC is also studied and the results indicated that the agglomeration of GO nanosheets largely inhibited the application of it in the AASC, even with a small dosage.

## 1. Introduction

Global warming and climate change related closely to carbon dioxide (CO_2_) emissions has sparked enormous concerns for the international community. It is well known that the construction industry is one of the major sources of sectorial CO_2_ emissions. Generally, the ratio of the production of ordinary portland cement (OPC) and CO_2_ emissions is about 1.0 [1]. Here, to reduce carbon footprints from the cement production, alkali activated slag composites (AASC) have been proposed as a kind of cement-free binder for construction use.

With the development of high-performance construction materials (HPCM) that possess excellent mechanical and multifunctional properties [2], the HPCM made from the AASC has also attracted tremendous attention. In particular, the addition of two-dimensional (2D) nanomaterials is one of the most effective reinforcing methods. Graphene (Gr) is a two-dimensional flat sheet with a thickness of one atom, possessing many advantages such as ultra-high specific surface area (2630 m^2^/g), excellent Young’s modulus (1 TPa), high electrical conductivity (2000 S/cm), superior ultimate strength (130 GPa), and high thermal conductivity of (5000 W/(m·K)) [3,4,5,6,7]. Currently, graphene oxide (GO) and reduced graphene oxide (rGO) are the two main graphene derivatives. Generally, both GO and rGO nanosheets have a certain amount of oxygen-containing functional groups [8], which can help these nanosheets to highly disperse in water solutions. In addition, rGO nanosheets have been also reported to uniformly disperse in the AASC matrix [9].

In terms of mechanical properties, the addition of 2D nanomaterials at a certain dosage can significantly increase the strength of the AASC [10]. Matalkah et al. [11] reported that the addition of 0.2 vol.% Gr nanosheets can improve the compressive strength of AASC by around 15%. Zhong et al. [12] indicated that the addition of 12 vol.% GO nanosheets can increase the compressive strength by about 50%. Saafi et al. [13] found that the flexural strength can be significantly enhanced by 134% with the addition of 0.35% wt. rGO. However, strength development of AASC containing the same nanosheets could exhibit a distinctly different trend due to different preparation procedures. Zhang et al. [14] found the maximum improvement of mechanical strength in AASC containing 3% wt. Gr nanosheets, while Ranjbar et al. [15] attributed this maximum improvement to the addition of 1% wt. Gr nanosheets. Liu et al. [16] and Zhu et al. [17] indicated that this maximum increment was owing to the addition of 0.3% wt. and 0.01% wt. GO nanosheets, respectively. Therefore, it is crucial to define the preparation procedures of nanosheets when investigating the strength development of the AASC. In our previous study [18], it can be found that rGO nanosheets prepared at 60 °C and 10 mol/L NaOH solution exhibited the highest increment of mechanical strength of the AASC compared to different temperatures (20, 40, and 80 °C). However, the effect of the content of rGO nanosheets on mechanical strength of the AASC under this preparation procedure is still unclear.

In terms of multifunctional properties, the addition of a 2D nanomaterial can largely increase the electrical conductivity of the AASC. Saafi et al. [19] declared the feasibility of using AASC blended with rGO nanosheets as self-sensing structures, since the addition of rGO nanosheets increased the electrical conductivity more than threefold. Moreover, there are other functional applications of construction materials, such as supercapacitors and electromagnetic interference structures that are related to capacitance and dielectric properties, respectively. However, although Giuri et al. [20] reported that GO nanosheets can modulate the capacitance and conductivity of poly (3,4 ethylenedioxythiophene) polystyrene sulfonate (PEDOT: PSS), more details about the electrical properties of AASC containing rGO and GO nanosheets are still scarce.

Although the HPCM has been endowed with various merits, the realization of high performance generally compromises its flowability and workability [21,22]. As a science of the flow of a material, the rheology of the fresh paste is crucial for achieving the desired properties (e.g., mechanical and functional properties) of the final product and its practical applications (e.g., 3D printing). Zhong et al. [12] reported that the addition of GO nanosheets can enhance the sensitivity of the viscosity of the AASC to the shear rate. When the content of GO nanosheets was 0, 4.59, and 5.66 vol.%, the yield stress of the AASC was almost 4, 2000, and 1000 Pa, respectively. Zhou et al. [23] further demonstrated that the addition of GO nanosheets can increase the viscosity of the AASC by about 20 times when the shear rate was 0.05 s^−1^, and the relationship between the viscosity and the shear rate transformed from a linear correlation to a non-linear one due to a decrease in the size of the GO nanosheet. However, as another graphene derivative, the effect of rGO nanosheets on rheological behaviors of the AASC is also unclear.

The rheological, mechanical, and electrical properties of the AASC containing rGO nanosheets (0.1, 0.2, and 0.3% wt.) were investigated in this study. Meanwhile, AASC containing GO nanosheets (0.1, 0.2, and 0.3% wt.) were added as reference groups. The rheometer was applied to characterize the rheological parameters of the fresh pastes including shear stress, yield stress, viscosity, and thixotropy. In particular, the compressive strength was used to illustrate the strength development of the AASC. Furthermore, electrochemical impedance spectroscopy (EIS) was performed to evaluate the electrical properties, dielectric properties, and electrical conductivity of the AASC. In general, this paper gives new insights into the material design of the HPCM made from the AASC and graphene derivatives.

## 2. Materials and Experiment Procedures

### 2.1. Raw Materials

Ground granulated blast furnace slag (GGBFS) used in this paper was purchased from Wuhan SinoCem Smartec Co., Ltd., Wuhan, China, confirming to the Chinese Standard GB/T 18046-2008. Figure 1 illustrates the particle size distribution of GGBFS. Additionally, the properties of GGBFS are shown in Table 1. The alkaline activator was made from a mixture of Na_2_SiO_3_ solution (27.8% wt. SiO_2_, 8.8% wt. Na_2_O, and 63.4% wt. H_2_O) and solid NaOH flakes (96% purity). Additionally, Table 2 lists the properties of the graphite oxide powder (purchased from Changzhou Sixth Element Materials Technology Co., Ltd., Changzhou, China).

### 2.2. Experiment Procedures

#### 2.2.1. Preparation of Nanomaterials

Graphite oxide powder was magnetically stirred with deionized water for 30 min to get graphite oxide solution. Subsequently, the resulting solution (16.7 g/L) was ultrasonized for 2 h at a power of 400 W and a frequency of 25 Hz. After ultrasonication, the well dispersed GO solution was obtained.

The partial GO solution was extracted and divided into three parts that contained 0.6, 1.2, and 1.8 g of solid GO nanosheets, labelled GO1, GO2, and GO3, respectively. According to our previous study [18], 100 g of NaOH solution (10 mol/L) was added to GO3 solution, followed by heating for 3 h at a temperature of 60 °C. After heating, the rGO solution containing 1.8 g of solid rGO nanosheets was obtained. For keeping the same reduction degree, 33 g and 67 g of NaOH solution (10 mol/L) were stirred with GO1 and GO2 solutions, respectively. After heating for 3 h at a temperature of 60 °C, the rGO solution containing 0.6 and 1.2 g of solid rGO nanosheets were also produced.

Meanwhile, GO and rGO solutions were produced through the same procedures, followed by two experiments. The structural defects in the GO and rGO nanosheets were detected by Raman scattering. Furthermore, the morphologies, especially the oxygen-containing functional groups of the GO and rGO nanosheets, were scanned by transmission electron microscopy (TEM).

#### 2.2.2. Sample Preparation

The mixing proportions of the samples are given in Table 3. The water-to-binder (w/b), Na_2_O-to-slag and SiO_2_-to-Na_2_O ratios were 0.5, 0.06, and 1.2, respectively. The rGO/GO-to-slag ratios included 0, 0.01, 0.02, and 0.03. The preparation of samples was illustrated as follows. After preparing GO and rGO solutions, residual NaOH flakes were mildly stirred with Na_2_SiO_3_ solution for 1 min to produce alkaline activator, followed by mixing with remainder water for 1 min. Then, GO/rGO solutions and alkali activator were homogeneously mixed for another 1 min. Finally, the obtained solution and slag were mixed for 4 min (2 min at a rotation speed of 62 ± 5 r/min and then another 2 min at a rotation speed of 125 ± 10 r/min). The obtained pastes were cast into two molds with dimensions of 20 × 20 × 20 mm^3^ and 40 × 40 × 40 mm^3^ and cured at a relative humidity >95% and a temperature of 20 ± 2 °C for 24 h. According to previous studies [18,24], alkali-activated composites can exhibit the highest mechanical strength after curing for 48 h at 80 °C owing to the completion of alkali activation. Therefore, a chamber with 80 °C was used to steam cure the demolded samples for 48 h.

#### 2.2.3. Rheological Tests

As a science of the deformation and flow of materials, rheology can describe the relationships between the stress, viscosity, and shear rate. In this study, the RM 100 touch device manufactured by Lamy Rheology Instruments (Lamy Rheology Instruments Company, Champagne-au-Mont-d’Or, France) was used to evaluate the rheological properties of the fresh pastes. The RM 100 touch device is shown in Figure 2. The temperature during the test was 25 °C and the measuring system used in this study was DIN-2. The paste was pre-sheared at 90 s^−1^ for 5 s, followed by a step-wise increase in shear rate from 5 to 90 s^−1^ in 10 steps with 5 s interval per step, as shown in Figure 3. Since the fast polycondensation reaction of alkali activated materials, the rheological behaviors of the paste were just measured at 5 min after mixing and the time-dependent behaviors were without consideration.

After testing, the plots of shear stress and shear rate can be obtained. Here, the Bing-ham model is used to fit the plot curves and describe the slope and intercept the linear relationship as the plastic viscosity (η_p_) and yield stress (τ_0_), as shown in the equation:(1)$$\tau =\tau_{0}+\eta_{p}\times \gamma$$
where η_p_ is the plastic viscosity (Pa∙s), τ is the shear stress (Pa), γ is the shear rate (s^−1^), and τ_0_ is the yield stress (Pa). In addition, the shade area consisting of the up and down curves in the plots of shear stress and shear rate was calculated to determine the thixotropy, which represents the structural changes of the flocs of the pastes, as shown in Figure 4.

#### 2.2.4. Mechanical Performance Test

A computerized universal testing machine (YHZ-300) with a loading rate of 2400 N/s was used to perform the compressive strength test, conforming to the requirements of Chinese Standard GB/T 17671-1999. The hardened samples (40 × 40 × 40 mm^3^) were used and the six tested values was averaged as the strength value of each group.

#### 2.2.5. X-ray Diffraction (XRD)

X-ray diffraction (XRD) was used to characterize the mineralogical composition of the samples by a DX 2500 for the purpose of the analysis of hydration products. The well-hydrated pastes were ground into uniform powders finer than 45 μm. The XRD experiment parameters were as follows: the test employed Cu *K*α radiation (λ = 1.54 Ǻ with a fixed divergence slit size of 0.5 °C and a rotating sample stage) at the range of 3° ≤ 2θ ≤ 80°, a scan rate of 0.02° per step, a voltage of 40 kV, and a current of 40 mA.

#### 2.2.6. Thermogravimetric Analysis (TGA)

The hydration product content is calculated by Thermogravimetric analysis (TGA), and the powdered paste samples were used in this test. The thermogravimetric (TG) and differential thermogravimetric (DTG) curves were acquired using the thermogravimetric analyzer on a STA 8000, SQ8T. The weight loss was monitored as follows: heating from 25 °C to 1050 °C, heating rate of 10 °C/min and purging under nitrogen atmosphere. The weight loss and quantified data were reported as mass percentage of dried pastes (at 30 °C).

#### 2.2.7. Electrochemical Impedance Spectroscopy (EIS)

As an advanced, non-destructive testing method, the EIS test is usually used to detect the impedance response of a material under an applied alternate current regime. In this study, the electrical behaviors of hardened samples with a dimension of 20 × 20 × 20 mm^3^ were tested using the electrochemical workstation (type PARSTAT 4000, AMETET Ltd., San Diego, CA, USA) under a sinusoidal potential perturbation of 100 mV and a frequency of 1 Hz to 1 MHz. The electrical spectrum of each group was obtained through the measurement of six samples from this group.

Meanwhile, the dielectric properties and electrical conductivity of a material can also be obtained by EIS test. In theory, the measurement of dielectric properties typically involves a parallel plate capacitor configuration, while the capacitance can be measured using the EIS test. It should be noted that an electrically insulating film (i.e., Teflon sheet) is needed when the test material possesses conductivity as the EIS test is not designed for measuring non-insulated materials. Previous studies [25,26] also reported the necessity of using electrically insulating films to ensure the accuracy of the EIS measurement for non-insulated materials. Therefore, the coupling of a sample and Teflon sheets was used for characterizing dielectric properties and electrical conductivity in this study, and each sheet was positioned between the sample and the copper electrode, as shown in Figure 5. The tests were operated under the parameters: frequency range of 1 Hz to 1 MHz and a sinusoidal potential perturbation of 1000 mV (i.e., 0.5 V/mm). Additionally, the electrical spectrum of each group was obtained through the measurement of six samples.

## 3. Results and Discussion

### 3.1. Characterization of GO and rGO

The characterization of well dispersed GO nanosheets and rGO nanosheets is clearly shown in Figure 6 and Figure 7. Raman spectra of the GO and rGO nanosheets is revealed in Figure 5. It illustrates that graphene-class nanomaterials possess three dominant peaks. The diamondoid mode leads to the presence of the first band, named the D-band, at 1380 cm^−1^. The second band, named the G-band, is at 1620 cm^−1^. Additionally, the second order of the D peak generates a third band as known as the 2D band at 2900 cm^−1^. According to previous studies, the strength ratio of the D-band (ID) to G-band (IG) can be used to predict the degree of defects in graphene-class nanosheets because the ratio is proportional to the structural defects in graphene-class nanosheets. In other words, more structural defect locations cause a higher ID/IG value [27,28]. As shown in Figure 6, the ID/IG value of rGO nanosheets increased from 0.88 to 0.92 compared to GO nanosheet, indicating that the rGO nanosheets contains structural defects after reduction.

Additionally, the TEM images of the nanosheets are shown in Figure 7, which confirms the irregular sheet-like structure of graphene-class nanomaterials. As shown in Figure 7a,b, the structural defects in nanosheets increased due to the alkaline reduction, which is consistent with the results of the Raman spectra. According to previous studies [29,30,31,32], the wrinkled and folded areas on the surface of the GO nanosheets can be attributed to the intercalation of the abundant oxygen-containing groups. Therefore, it can be clearly observed from Figure 7b–d that the amount of oxygen-containing groups on rGO nanosheets decreased compared to GO nanosheets due to the alkaline reduction. It can also be demonstrated that the NaOH solution (10 mol/L, 60 °C) can effectively reduce the GO nanosheet.

### 3.2. Rheological Properties

Figure 8 exhibits the flow curves of the samples and the linear curves fitted by the Bingham model. It can be concluded that the shear stress of the samples enhanced with the shear rate independent of the rGO/GO content. Additionally, the increment on the shear stress of the samples enhanced with the increasing rGO/GO content. In addition, the details of the fitting curves can be found in Table 4. The R^2^ values of all flow curves were higher than 0.99, indicating that the rheological behaviors of the AASC containing rGO/GO nanosheets tightly conformed to the Bingham model.

Figure 9 shows the variations in rGO and GO contents in the yield stress and plastic viscosity of the samples. With the increasing rGO/GO content, the yield stress of the samples gradually increased. Compared to R0, the yield stress of rG1, rG2, and rG3 increased by 24.80, 75.93, and 186.77 times, respectively. A similar trend can be found in the plastic viscosity of the samples. Compared to R0, the plastic viscosity of G1, G2, and G3 increased by 1.08, 1.71, and 2.66 times, respectively. Generally, yield stress represents the maximum stress that prevents the plastic deformation and viscosity represents the resistance to flow and refers to the slope of the Bingham curve. Therefore, it can be concluded that the deformation of the samples becomes more difficult under the external force when the rGO/GO content increases. These changes can be explained by the following reasons. Firstly, nanosheets can fill the gaps between particles, thus increasing the force of friction among particles. Secondly, nanosheets can accelerate the polycondensation reaction of aluminosilicate particles in the alkaline solution and further increase the integrity of the samples [18]. According to Zhou et al. [23], the integrity of the complex liquid is proportional to the viscosity. Thirdly, 2D nanomaterials can absorb a large amount of water inside their interlayers due to an excellent specific surface area [33], leading to the reduction of the content of the lubricant among the particles.

Furthermore, the plastic viscosity and yield stress of the samples with rGO addition were always higher than those with GO addition under the same content, as shown in Figure 9. Compared to G3, the plastic viscosity and yield stress of rG3 increased from 8.555 to 13.074 Pa and from 0.910 to 1.258 Pa∙s, respectively. This phenomenon could be due to the higher dispersion degree of rGO nanosheets in the alkaline solution, leading to more water restrained into the interlayers. In addition, the thixotropy values of the samples are listed in Table 4. Compared to R0, the thixotropy value of rG3 and G3 increased from 51.08 to 773.75 and 424.00 Pa/s, respectively. This indicates that there are abundant flocculation structures in the AASC due to the addition of rGO and GO nanosheets.

### 3.3. Compressive Strength

Figure 10 shows the strength development of the samples containing different rGO/GO contents. As can be seen from the figure, the compressive strength of R0 under the w/b of 0.5 is about 37.1 MPa.

It can be detected that the increasing rGO content significantly improved the compressive strength of the samples. Compared to R0, the strength of rG1, rG2, and rG3 increased from 37.1 to 40.6, 42.0, and 44.9 MPa, respectively. In particular, the addition of 0.3% wt. rGO nanosheets increased the compressive strength of the AASC by 21.02%. This phenomenon can be explained by the filler effect on the microstructure and the acceleration effect on the polycondensation reaction of the slag, which can lead to a denser matrix and more hydration products, therefore increasing the mechanical properties of the AASC. Furthermore, the samples with rGO addition can simultaneously generate C-S-H(I) and C-A-S-H gels [18], and the gaps between unhydrated particles and different phases were bridged, which leads to superior compressive strength of the AASC [34,35].

As illustrated in Figure 10, the compressive strength of G1 and G2 increased from 37.1 to 41.4 and 43.3 MPa compared to R0. However, the addition of 0.3% wt. GO nanosheets reduced the strength of the samples to 27.2 MPa, which can be attributed to the massive agglomeration of GO nanosheets in the highly alkaline matrix. Agglomerated GO nanosheets can form defects in the microstructure, facilitating the crack expansion. Previous studies [36,37,38,39] have also reported similar experimental results that GO nanosheets can be agglomerated in an alkaline solution. It should be noted that the agglomeration of GO nanosheets cannot be detected in the rheological test, indicating that GO nanosheets may begin to agglomerate on a large scale only after the matrix had undergone a certain degree of the polycondensation reaction. In addition, the reinforcing effect of GO nanosheets was more obvious than rGO nanosheets when the content was less than 0.2% wt., owing to the more oxygen-containing groups and less structural defects of GO nanosheets.

### 3.4. X-ray Diffraction (XRD) Analysis

The XRD patterns of the samples are determined to identify the microscopic products of the alkali activated materials with rGO/GO addition, as shown in Figure 11. The patterns reported that the GO/rGO-reinforced alkali activated materials have two crystalline hydration products. These results indicated that the chemical composition of the alkali activated materials was not changed with GO/GO addition but changed the hydration process. The peak at 2θ = 29.27° was C-A-S-H phase (PDF #00-033-0306), and the crystalline C-S-H (I) (C-S-H with low Ca/Si ratio, PDF #00-034-0002) was detected for the samples with rGO addition, which featured a high content of Si present as a Si-rich hydrate. Additionally, the XRD patterns of rGO/GO-reinforced alkali activated materials are similar to the previous research done by Long et al. [18].

Figure 11 shows that the C-A-S-H intensities in R0 were lower than that in G1 and G2, in which the GO addition increased from 0.1% wt. to 0.2% wt. On the contrary, when the GO content increased from 0.2% wt. to 0.3% wt., the crystalline C-A-S-H phase exhibits the opposite phenomenon as a result of the massive agglomeration of GO nanosheets in the highly alkaline matrix. These results showed that when the GO content is less than 0.2% wt., there will be a promotion effect for the hydration process, confirming the analysis of the compressive test.

In addition, the strength peak of C-S-H (I) in the alkali activated materials containing rGO is higher than that in the alkali activated materials containing GO, indicating that the alkali activated materials reinforced by rGO preferred to generate C-A-S-H and C-S-H(I) phases. This is due to the low content of Al and the high content of Si in the vicinity of the rGO nanosheet.

### 3.5. Thermogravimetric Analysis (TGA) Analysis

DTG results of the alkali activated materials with GO/rGO addition are depicted in Figure 12. The figure reveals that the main mass loss is in the temperature range of 50–200 °C of the samples related to the C-A-S-H gels [40,41]. However, the dehydration peak of C-S-H (I) was not obvious in the DTG analysis, likely due to the amorphous structure of C-S-H (I).

According to Figure 12a, the peak of C-A-S-H gels in alkali activated materials containing 0.2% GO had an obvious increment. However, the percentage of the gels was clearly reduced when the GO content contained 0.3%. It is in accord with the XRD and compressive strength analyses that the agglomerated GO nanosheets could inhibit the formation of the C-A-S-H gels. Additionally, it can be seen from Figure 12b that the C-A-S-H gel peak of rG3 is obviously higher than G3, and the hydration products is related to the mechanical strength of the samples. Therefore, it can be concluded that when the nanomaterials dosage is 0.3% wt., the reinforcing effect of rGO nanosheets is more obvious than the GO nanosheets from the micro-product perspective analysis. Additionally, the analysis corresponded to the results of the compressive strength test.

### 3.6. Electrical Properties

Figure 13 shows the impedance responses of the samples with rGO and GO addition. Table 5 lists the impedance values of the samples at the specific frequencies. As for all samples, the impedance diagram consists of a semicircle and a line, representing a resistive behavior at low frequencies and a capacitive behavior at high frequencies [42]. Moreover, the impedance of the geopolymer pastes is generally higher than that of the OPC pastes [33]. It can be clearly seen from Figure 13a that the diameter of the semicircle of rG1 and rG2 increased gradually compared to R0, revealing that rGO nanosheets can improve the impedance of AASC. It can be attributed to the acceleration effect of rGO on the polycondensation reaction of the AASC, leading to an increment on the compactness of the matrix. According to Long et al. [29], a denser matrix generally possesses a higher impedance value. It should be noted that the diameter of the semicircle of rG3 significantly decreased, indicating that the addition of 0.3% wt. rGO nanosheets can decrease the impedance of the AASC. This phenomenon can be attributed to excellent electrical conductivity of well-dispersed rGO nanosheets from the exfoliation of oxygen-containing groups. In particular, the reduction in the impedance of the samples cannot be found in rG1 and rG2, since the conductivity provided by small dosages of rGO nanosheets was unable to neutralize the acceleration effect on the polycondensation reaction. A similar trend can be detected in the samples with GO addition, as shown in Figure 13b. Compared to R0, the impedance of G1 and G2 increased due to the motivation on the polycondensation reaction. However, the mechanism of the reduction in the impedance of G3 was different with that of rG3. The former can be explained by more defects in the matrix derived from the massive agglomeration of GO nanosheets, facilitating the transportation of the conductive ions in pore solutions (mainly Na^+^ and OH^−^).

Bode and Nyquist curves of the samples with coupling graphene derivatives and electrically insulating films are depicted in Figure 14 and Figure 15, respectively. Compared to the samples with no film, the samples with films exhibited the same tendency that the impedance responses increased and then decreased with the increasing rGO/GO content. Moreover, it can be observed that the impedance of the samples amplified by several times due to the incorporation of electrically insulating films. In particular, the impedance of the samples enlarged more than threefold in the low frequency region. According to previous studies [25,26], the measurement of the electrical properties (especially dielectric properties) of the samples with films may be more accurate, as the EIS technology is not designed for measuring conductive materials. Therefore, the Bode and Nyquist curves of the samples were used to analyze the effect of different graphene derivatives on the impedance responses of the samples. Obviously, the samples containing GO nanosheets had a higher impedance than rGO when the content was same or less than 0.3% wt. Like the strength analysis, this phenomenon could be due to the more oxygen-containing groups and fewer structural defects of GO nanosheets. In addition, the impedance of G3 decreased sharply due to the massive agglomeration of GO nanosheets. However, this plunge cannot be seen in rG3, verifying that the agglomeration was not responsible for the reduction in the impedance of rG3. It is consistent with the findings of a previous study [9], indicating that rGO nanosheets can uniformly disperse in the AASC matrix.

To evaluate the dielectric properties of AASC with graphene derivatives, the complex dielectric constant (κ), namely relative complex permittivity (*ε*r), was proposed. Generally, the κ is composed of a real part (*ε*′) and an imaginary part (*ε*″), which describes the interaction of a material within an electric field. More precisely, the real part (dielectric constant, *ε*′) represents the energy storage and the imaginary part (*ε*″) relates to the energy loss [2]. According to Roggero et al. [42], the experimental impedance spectra (*Z*_*im*_, *Z*_*re*_) can mathematically transform *ε*′ to *ε*″ using Equations (2)–(6):(2)$$\left|Z\right|=\sqrt{Z_{re}^{2}+Z_{im}^{2}}$$
(3)$$\varepsilon^{\prime }=\frac{Z_{im}}{\left(2\pi f\right)C_{v}\left(Z_{re}^{2}+Z_{im}^{2}\right)}$$
(4)$$\varepsilon^{\prime \prime }=\frac{Z_{re}}{\left(2\pi f\right)C_{v}\left(Z_{re}^{2}+Z_{im}^{2}\right)}$$
(5)$$C_{v}=\frac{\varepsilon_{v}A}{l}$$
(6)$$\sigma_{ac}=\left(2\pi f\right)\varepsilon_{v}\varepsilon^{\prime \prime }$$
where *f* is the frequency of the applied voltage, *ε_v_* is the vacuum permittivity (F/cm), *A* is the area of a tested sample (20 × 20 mm^2^), l represents thickness of samples (20 mm), *C*_*v*_ is vacuum capacitance, and *σ_ac_* is AC conductivity.

Figure 16 and Figure 17 show the evolution of the *ε*′ and *ε*″ of the samples with rGO/GO addition against frequency. Table 6 lists the values of the *ε*′ and *ε*″ of the samples at the specific frequencies. Compared to the results of previous studies [2,43], it can be concluded that the *ε*′ of alkali activated materials is usually higher than that of the OPC. As can be clearly observed from Figure 16 and Figure 17, the *ε*′ and *ε*″ of the samples decreased when the rGO/GO content increased from 0 to 0.2% wt., which is different from previous studies [2,44,45] that reported the feasibility of using rGO nanosheets as a dielectric reinforcement method. However, most of these studies focused on the utilization of a large dosage of rGO nanosheets rather than a small dosage (especially less than 1% wt.). For example, Phrompet et al. [2] found that the *ε*′ of C_3_AH_6_ cement can be significantly improved when the rGO content was 4% wt. In this study, this reduction can be attributed to abundant interlayer water restrained by rGO/GO nanosheet. Due to the relatively low complex permittivity of water, the presence of a large amount of the interlayer water can surpass the reinforcing effect of a dense matrix on the dielectric properties. Moreover, the *ε*′ and *ε*″ of the samples rose when the GO/rGO content increased from 0.2 to 0.3% wt. The former can be explained by the reduction in the amount of the interlayer water due to the massive agglomeration of GO nanosheets, which can impair the shielding effect of water on the increase of the *ε*′ and *ε*″. The latter indicated the appearance of the inflection point at around 0.2% wt. In other words, when the content was higher than 0.2% wt. and gradually increased, the *ε*′ and *ε*″ of the samples containing rGO nanosheets also gradually increases and finally exceeds that of R0.

The Maxwell–Weigner model can interpret this phenomenon, which is related to the interface polarizations of the conductor–insulator interface [2]. The model describes a dielectric medium with abundant boundaries between conducting grains (i.e., rGO nanosheet) and poor conducting grains (i.e., the matrix). Generally, when dielectric materials are positioned in an external electric field, the charge carriers can stay at the grain boundaries (particularly the oxygen vacancies and defects) to generate the polarizations and the dielectric constant [2]. Therefore, when the content of well-dispersed rGO nanosheets increased to 0.3% wt., rGO nanosheets can provide more storage sites for charge carriers. The increment on storage sites gradually overcame the shielding effect of the interlayer water, thus increasing the *ε*′ and *ε*″ of the samples. It can also explain why large doses of rGO nanosheets are usually chosen for enhancing the *ε*′ and *ε*″. It should be noted that rGO nanosheets from different preparation procedures possess different reduction degrees and structural defects. Consequently, in addition to the increment in rGO content, rGO nanosheets with higher reduction degrees or more structural defects can also be expected to supply more reinforcing effects.

It can be also seen from Figure 16 and Figure 17 that the points interwove each other in the ultra-high frequency area (>105 Hz), indicating that the voltage of 0.5 V/mm was not enough to support the measurement in this area due to the use of electrically insulating films. In addition, although the points still had a certain degree of disorder, a similar tendency can still be detected in the high frequency area (103–105 Hz). The *ε*′ and *ε*″ of the samples under high frequency area decreased and then increased when the rGO/GO content increased from 0.2 to 0.3% wt.

In addition, the AC conductivity (*σ_ac_*) can be also mathematically calculated using the electrochemical parameters from the EIS test, as shown in Equation (6). More details about the σac of the samples with the rGO/GO addition at 20, 100, 1000, and 100,000 Hz are shown in Figure 18 and Table 7. The σac of the samples decreased and then increased with the increasing rGO/GO content. Additionally, the σac of the samples with GO addition varied more obviously than that with rGO addition, especially the rGO/GO content at 0.3% wt. In particular, the increment of the σac of rG3 and G3 can be attributed to the emergence of the inflection point and the agglomeration, respectively.

From the above results, the AASC containing rGO/GO nanosheets has been evaluated electrically for applications including energy storage structures (related to *ε*′, e.g., supercapacitors and electromagnetic shielding structures) and energy loss structures (related to *ε*″ and σac, e.g., self-sensing structures). The well-dispersed rGO nanosheets with higher content or higher reduction degree has been proven to benefit these applications.

## 4. Conclusions

This study investigated the rheological, mechanical, and electrical properties of the AASC with rGO/GO addition (0, 0.1, 0.2, and 0.3% wt.). These conclusions can be drawn:(a)Both rGO and GO nanosheets can increase the shear stress, yield stress, plastic viscosity, and thixotropy of AASC. Under the same content, rGO nanosheets provided a higher improvement. In particular, 0.3% wt. rGO addition significantly increased the stress, viscosity, and thixotropy by 186.77 times, 3.68 times, and 15.15 times, respectively.(b)Both rGO and GO nanosheets can improve the compressive strength at 0.2 wt. and GO nanosheets made for better reinforcement. However, the addition of 0.3% wt. nanosheets exhibited a different trend, where rGO nanosheets increased the strength by 21.02% and GO nanosheets decreased it by 26.68%.(c)Independent of the use of insulating films, the impedance of the AASC with rGO/GO addition increased at 0.2% wt. and then decreased at 0.3% wt. The reduction can be attributed to an increasing amount of conductive rGO nanosheets and the agglomeration of GO nanosheets.(d)Both rGO and GO nanosheets can decrease the *ε*′ and *ε*″ of the AASC due to the presence of the interlayer water, when the content increased from 0 to 0.2% wt. However, the *ε*′ and *ε*″ of the AASC containing 0.3% wt. rGO nanosheets rose since rGO nanosheets provided abundant structural defects as the storage sites for charge carriers.(e)With the increasing of rGO/GO content, the σac of AASC decreased first and then increased. The inflection point of the *ε*′, *ε*″, and σac of the AASC with rGO addition can be found at around 0.2% wt. Therefore, it suggests that multifunctional applications of the AASC can be achieved by increasing the content and the reduction degree of the rGO nanosheets.

## Figures and Tables

**Figure 1 materials-14-04374-f001:**
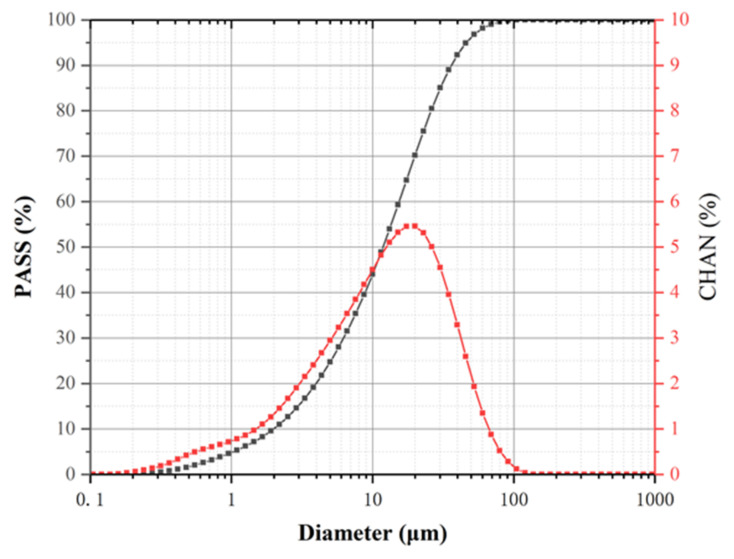
GGBFS’s particle size distribution.

**Figure 2 materials-14-04374-f002:**
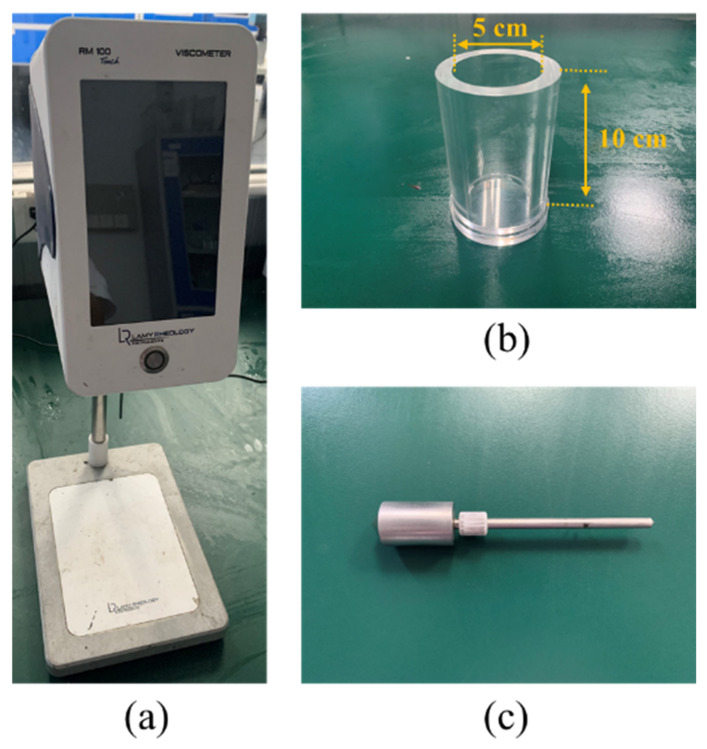
RM 100 touch device: (**a**) body of RM 100 touch device, (**b**) paste container, (**c**) standard tube.

**Figure 3 materials-14-04374-f003:**
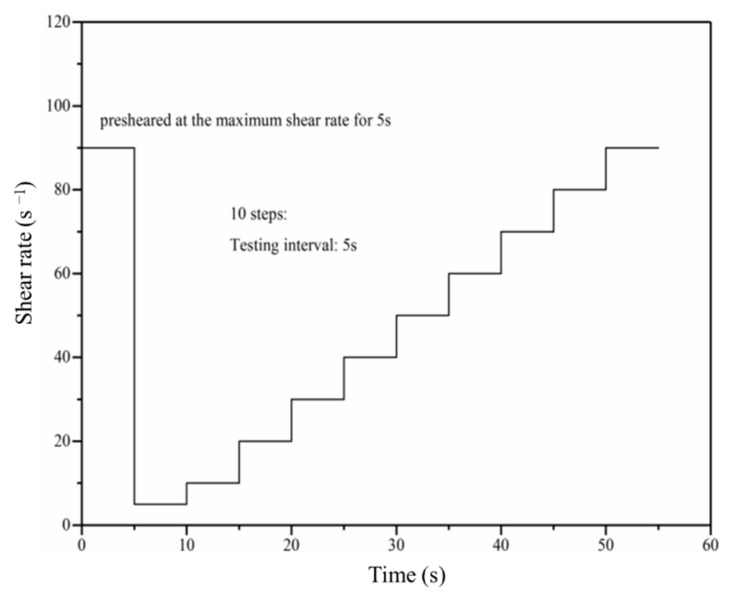
Rheological testing protocol of samples after mixing for 5 min.

**Figure 4 materials-14-04374-f004:**
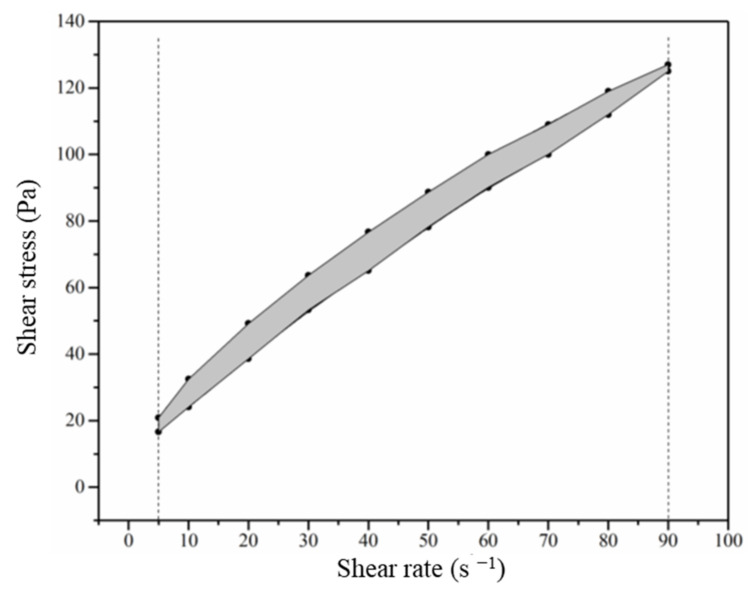
Thixotropy of the paste calculated from the shade area.

**Figure 5 materials-14-04374-f005:**
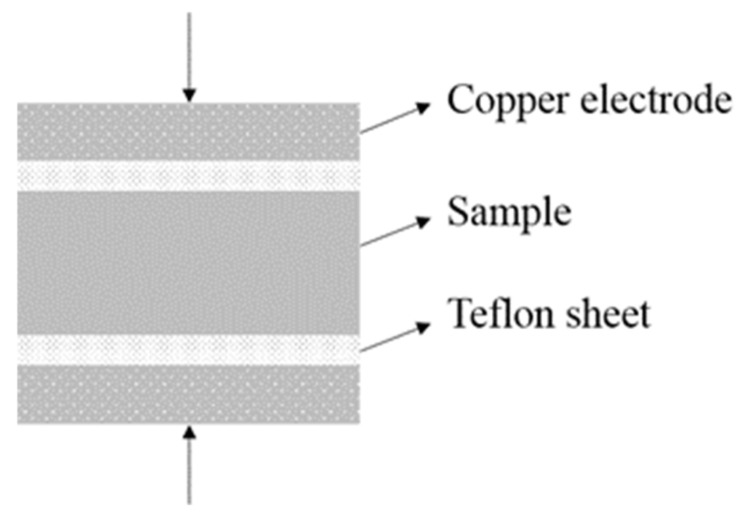
Test model of the samples for dielectric behaviors and electrical conductivity.

**Figure 6 materials-14-04374-f006:**
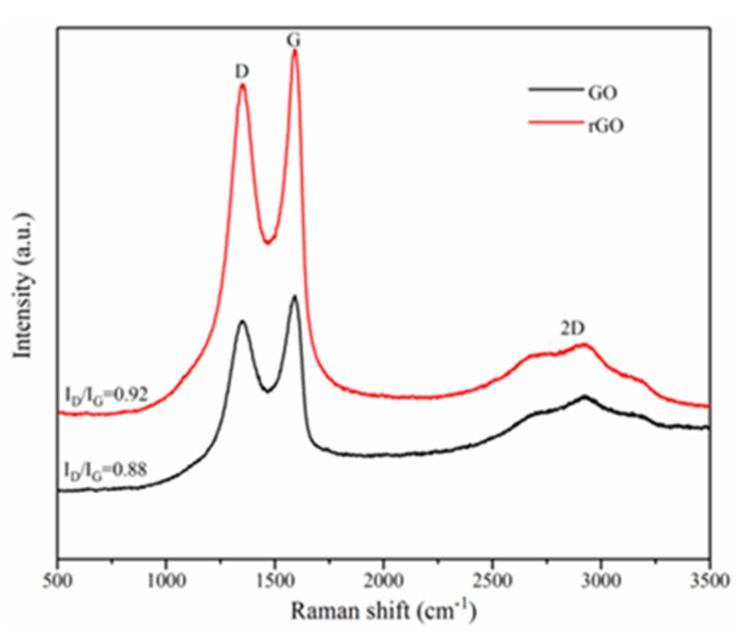
Test model of the samples for dielectric behaviors and electrical conductivity.

**Figure 7 materials-14-04374-f007:**
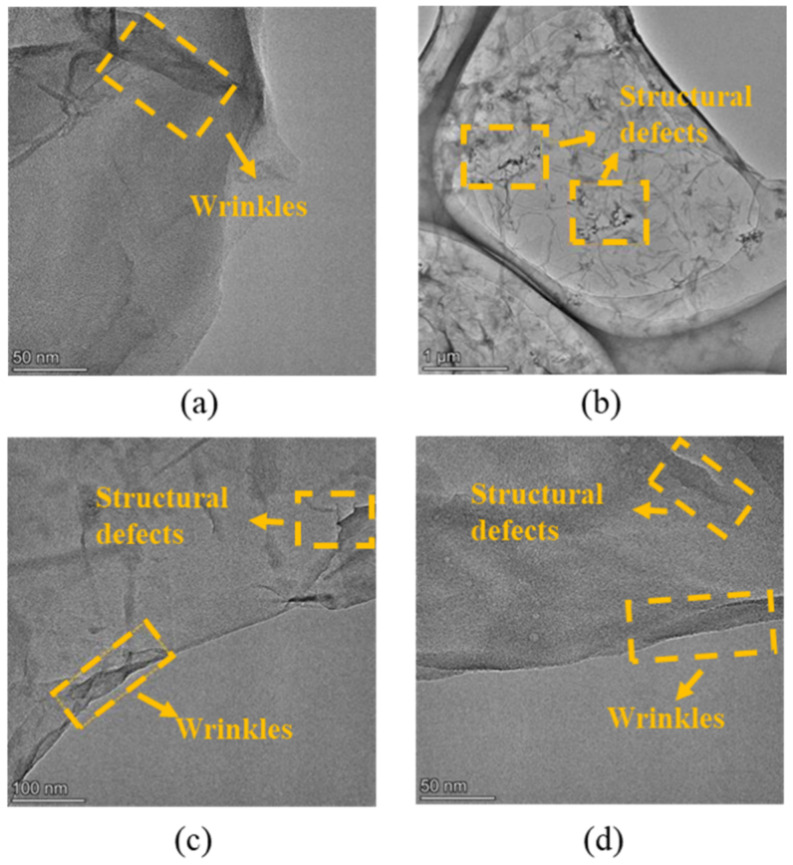
TEM images: (**a**) GO; (**b**–**d**) rGO nanosheet.

**Figure 8 materials-14-04374-f008:**
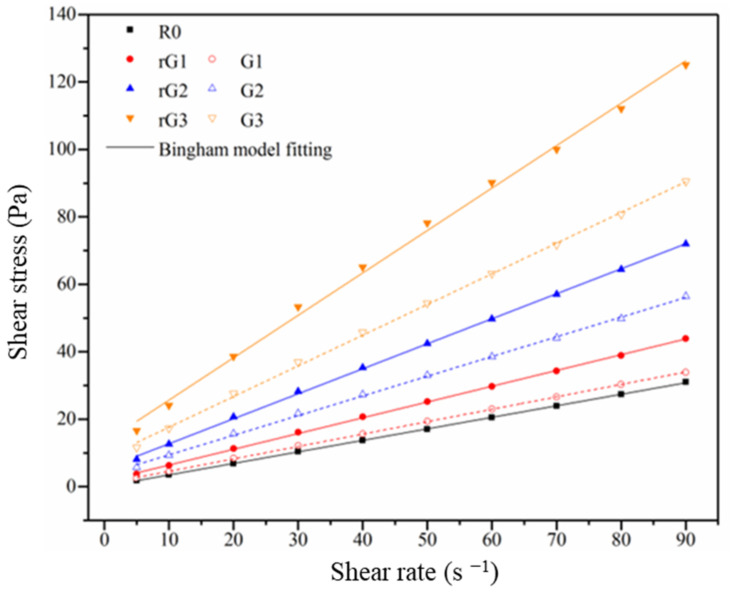
Flow curves of the samples.

**Figure 9 materials-14-04374-f009:**
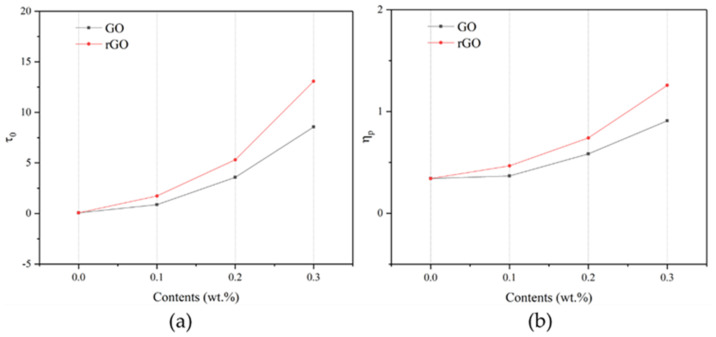
Variations between rGO and GO samples’ yield stress (**a**) and plastic viscosity (**b**).

**Figure 10 materials-14-04374-f010:**
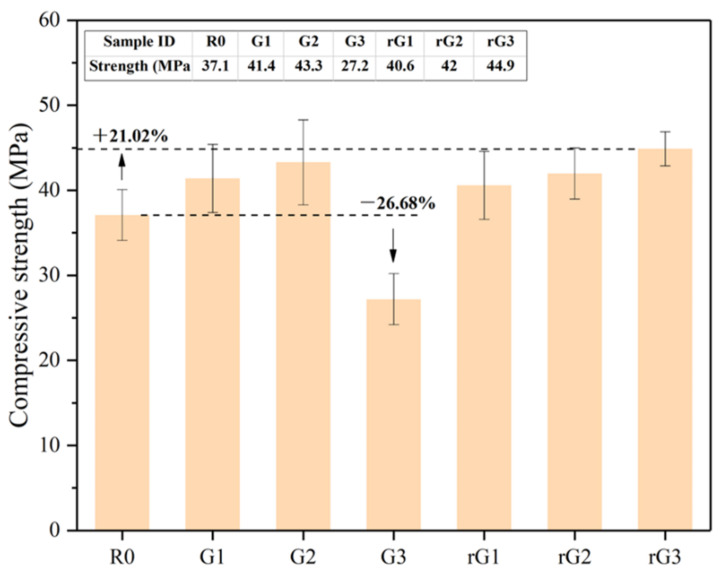
Compressive strength of the samples with rGO/GO addition.

**Figure 11 materials-14-04374-f011:**
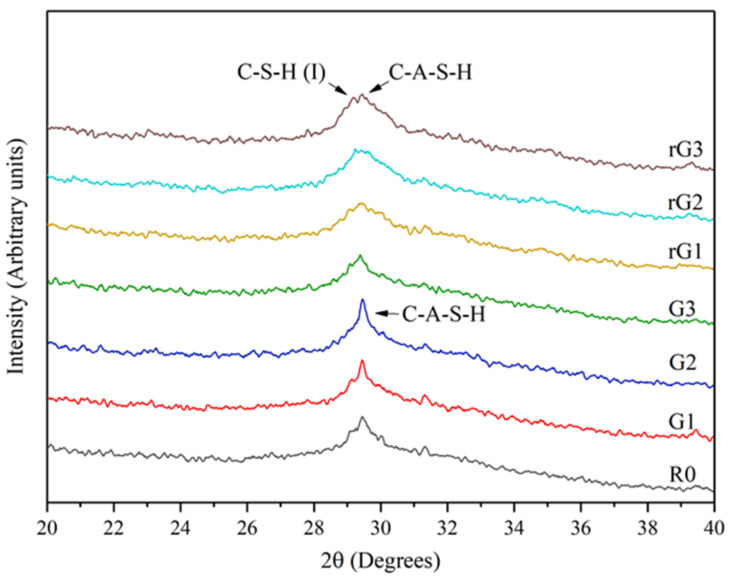
XRD patterns of the samples with GO/rGO addition.

**Figure 12 materials-14-04374-f012:**
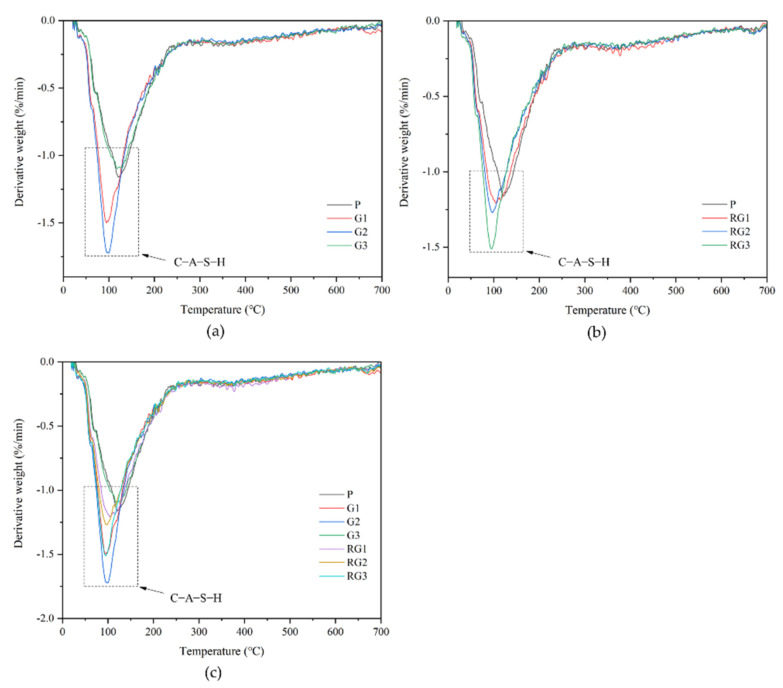
TGA results of the samples: (**a**) with GO addition; (**b**) with rGO addition; (**c**) with GO/rGO addition.

**Figure 13 materials-14-04374-f013:**
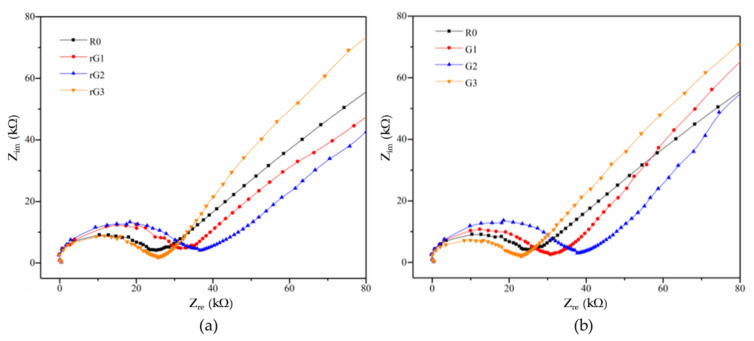
Impedance responses of the samples with rGO (**a**) and GO (**b**) addition.

**Figure 14 materials-14-04374-f014:**
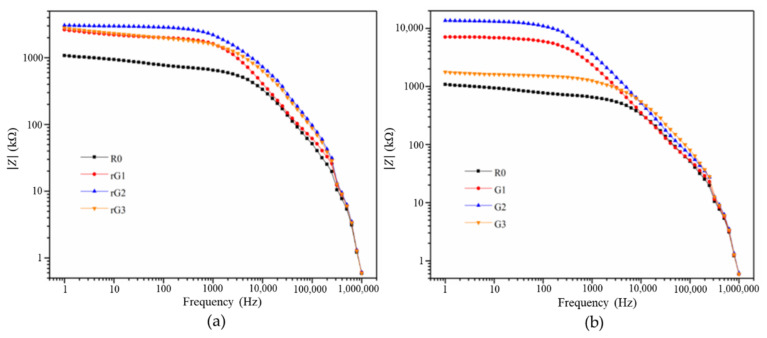
Bode curves of the samples with rGO (**a**) and GO (**b**) addition.

**Figure 15 materials-14-04374-f015:**
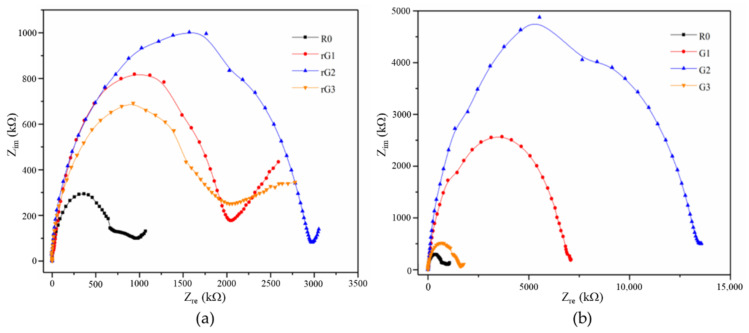
Nyquist curves of the samples with rGO (**a**) and GO (**b**) addition.

**Figure 16 materials-14-04374-f016:**
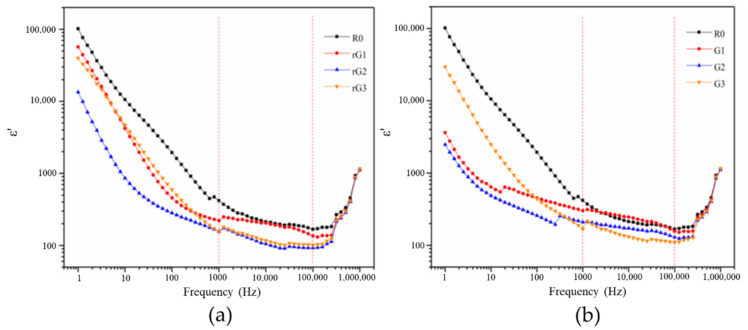
Dielectric constant of the samples with rGO (**a**) and GO (**b**) addition.

**Figure 17 materials-14-04374-f017:**
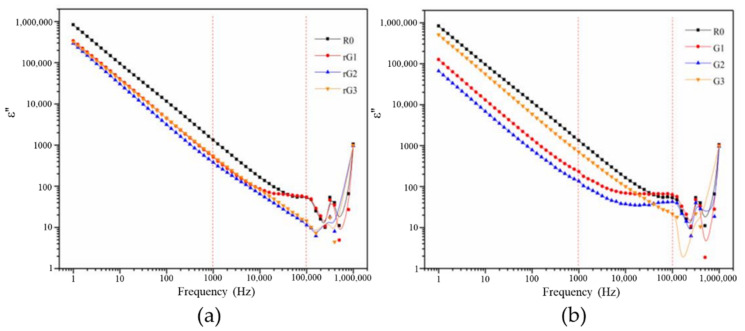
Dielectric loss of the samples with rGO (**a**) and GO (**b**) addition.

**Figure 18 materials-14-04374-f018:**
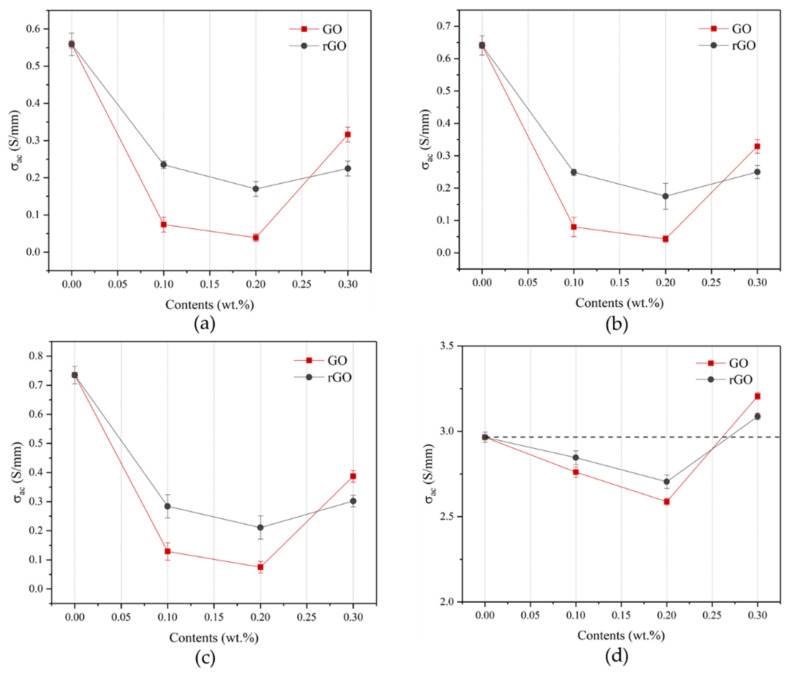
Electrical conductivity of the samples with GO and rGO: (**a**) 20 Hz; (**b**) 100 Hz; (**c**) 1000 Hz; (**d**) 100,000 Hz.

**Table 1 materials-14-04374-t001:** Properties of the GGBFS.

Component (wt.%)		Slag
Physical properties	Bulk density (g/cm^3^)	1.20
	Density	2.86
	Specific surface area (m^2^/kg)	431.00
Chemical compositions	CaO	39.81
	SiO_2_	31.86
	Al_2_O_3_	16.53
	MgO	6.89
	K_2_O	0.54
	Na_2_O	0.33
	MnO	0.18
	Fe_2_O_3_	0.43
	TiO_2_	1.23
	Loss on ignition	0.02

**Table 2 materials-14-04374-t002:** Graphite oxide properties.

Appearance	Solid Graphite Oxide Content (Mass%)	pH	Carbon Content in Solid Graphite Oxide (Mass%)	Sulfur (Mass%)	Chloride (Mass%)
Black paste	43.57	2.21	50.45	0.47	0.8

**Table 3 materials-14-04374-t003:** Mixing proportions of samples.

Sample ID	Slag	Na_2_O	SiO_2_	Water	rGO	GO
R0	600	36	43.2	300	-	-
G1	600	36	43.2	300	-	0.6
G2	600	36	43.2	300	-	1.2
G3	600	36	43.2	300	-	1.8
rG1	600	36	43.2	300	0.6	-
rG2	600	36	43.2	300	1.2	-
rG3	600	36	43.2	300	1.8	-

**Table 4 materials-14-04374-t004:** Parameters of fitting curves using the Bingham model.

Sample ID	η_p_ (Pa.s)	τ_0_ (Pa)	R^2^	Thixotropy (Pa/s)
R0	0.342	0.070	0.999	51.08
G1	0.368	0.881	0.999	73.30
G2	0.584	3.584	0.999	171.73
G3	0.910	8.555	0.999	424.00
rG1	0.467	1.736	0.999	130.53
rG2	0.741	5.315	0.999	308.60
rG3	1.258	13.074	0.999	773.75

**Table 5 materials-14-04374-t005:** Impedance values of the samples at specific frequencies (kΩ).

Sample ID	20 Hz		100 Hz		1000 Hz		100,000 Hz	
	*Z* _*re*_	*Z* _*im*_	*Z* _*re*_	*Z* _*im*_	*Z* _*re*_	*Z* _*im*_	*Z* _*re*_	*Z* _*im*_
R0	81.45	56.83	48.21	25.06	31.49	7.85	18.00	8.43
rG1	97.51	62.29	58.26	29.57	38.24	8.85	19.96	12.49
rG2	61.54	24.25	47.84	10.99	38.88	4.97	18.33	13.33
rG3	35.70	13.79	30.10	5.25	26.51	2.20	15.82	8.61
G1	44.61	16.38	37.32	6.72	32.13	3.11	19.08	9.90
G2	53.47	16.18	45.22	7.36	39.32	3.46	18.55	13.68
G3	37.04	18.58	28.65	7.19	24.05	2.60	13.30	7.14

**Table 6 materials-14-04374-t006:** Dielectric constant and loss of the samples at specific frequencies.

Sample ID	20 Hz		100 Hz		1000 Hz		100,000 Hz	
	*ε*′	*ε*″	*ε*′	*ε*″	*ε*′	*ε*″	*ε*′	*ε*″
R0	6373.32	50,321.53	1935.90	11519.5	416.85	1321.72	167.37	53.32
rG1	1959.22	21,086.24	458.24	4485.30	219.71	510.50	135.88	51.19
rG2	527.90	15,307.92	281.12	3139.58	156.00	379.31	92.13	48.64
rG3	2460.38	20,258.68	592.72	4492.63	154.19	543.60	101.30	55.51
G1	638.53	6664.85	451.39	1440.24	301.23	231.71	156.64	49.65
G2	391.64	3489.11	257.37	779.66	208.28	134.43	129.28	46.54
G3	1362.76	28,430.30	439.26	5909.84	168.30	696.64	110.10	57.66

**Table 7 materials-14-04374-t007:** Electrical conductivity of the samples at specific frequencies (S/mm, ×10^−3^).

Sample ID	20 Hz	100 Hz	1000 Hz	100,000 Hz
R0	0.559	0.641	0.735	2.965
rG1	0.235	0.249	0.284	2.846
rG2	0.170	0.175	0.211	2.705
rG3	0.225	0.250	0.302	3.087
G1	0.074	0.080	0.129	2.761
G2	0.039	0.043	0.075	2.588
G3	0.316	0.329	0.387	3.206

## Data Availability

Data sharing is not applicable to this article.

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
