# Peer review of "Investigation of Graphene Derivatives on Electrical Properties of Alkali Activated Slag Composites"

_materials, 2021, doi:10.3390/ma14164374_

Round 1
Reviewer 1 Report
The manuscript falls on the aim and scope of the journal. The experimental and discussion area sufficient except some areas.
Need to check minor grammar and typos.
Author Response
Dear Alexander Code (MPhys) editor:
Many thanks for your email dated July 27, 2021. Based on the comments of the reviewers, the manuscript has been revised accordingly. We sincerely appreciate for reviewers’ careful reading and valuable comments to improve the quality of this paper. The revised parts are highlighted in the attached manuscript. In addition, we would like to provide the following responses to the reviewers’ comments.
Reviewer 1
Comments and Suggestions for Authors:
The manuscript falls on the aim and scope of the journal. The experimental and discussion area sufficient except some areas.
Need to check minor grammar and typos.
Response: Thank you for your recognition and good comments. We sincerely appreciate for your careful reading and valuable comments. The comments are very helpful for revising and improving the paper. We have studied comments carefully and have made correction which we hope meet with approval. The main corrections in the paper can be seen from the revised manuscript, and the revised parts are highlighted in the attached manuscript.
Reviewer 2 Report
- The subject is suitable for the journal and is relevant to current interests.
- English should be improved.
- Using word “electrochemical” is not appropriate within the whole paper. Electrochemistry refers to a relationship between electrical potential, as a measurable and quantitative phenomenon, and identifiable chemical change, with either electrical potential as an outcome of a particular chemical change, or vice versa. Since there is no chemical change associated with this investigation, the expression “electrical properties” must be used instead, even in the title.
- The abbreviated labelling of the graphene oxide in the abstract is bit confusing. Once it is assigned as rGO, the second time only as GO. The same problem occurs in section 2.1.1. It is not clear whether GO refers to graphene oxide or graphite oxide. Unify the labelling or change the description!
- Concerning table 2, it is strange that carbon content is much higher than the total solid content. Can you explain this discrepancy?
- How was the specific surface area of GGBS determined? Please indicate the method (Blaine, BET).
- What is the nature of graphene oxide? You are talking about powder, but table 2 indicates the appearance as black paste and solid content is only 43%. So, is it more like an aqueous suspension?
- The pH of graphite oxide is quite low. Was this taken into account in the composition of the AASC mixtures? rGO is neutralised during preparation by the addition of NaOH but the GO solution remained strongly acidic. In that case you have different mixture proportions of the AAS matrix between R0, G* and rG* samples.
- Data for the rG3 mixture are missing in Table 3 and Table 4.
- What was the temperature and device geometry used during rheological tests? What is the wavelength of XRD source? Description of the methodology is not sufficient.
- Specify the title of 3.1. subsection!
- The resolution of TEM images in Fig. 6 must be improved. The labels in the micrographs are not legible even at high magnification.
- Row 251 – AASC cannot be regarded as Newtonian fluid because Bingham model describes non-Newtonian behaviour (it possesses a yield stress). However, if we take into account a measurement error, the yield stress of the samples R0 and G1 is so low that it can be considered a zero. Therefore, the Newtonian behaviour could be practically attributed only to these two mixtures.
- Rows 262–264 – probably tossed meaning of the sentence because yield stress represents the maximum stress that prevent the plastic deformation and viscosity represents the resistance to flow and refers to the slope of the Bingham curve.
- Can author provide any reference to the statement concerning the rows 342–345? Moreover, this sentence makes no sense because it is grammatically incorrect.
- Rows 355–356 – Please, explain in more detail how the thermogravimetric measurement explains the reinforcing effect of rGO nanosheet compared to GO nanosheet. In my opinion, thermogravimetry provides no information about the morphology or the mechanical properties.
- Row 372 (and further on) – replace the word “polymerization” rather with “polycondensation”. Polymerization is a radical reaction which is associated with chaining of organic alkenes. Moreover, the molecular structure of C-A-S-H and C-S-H gels is not polymeric but rather oligomeric.
- The Zim should be positive. In my opinion, it refers to the capacitive reactance which has always a positive value. If it was negative, then it would refer to inductance and this is not this case. Change the sign!
- Equations 2, 3, and 4 are incorrect. The square root is missing in Eq. 2. The negative sign before Z (Eq. 3 and 4) must be omitted because dielectric constant cannot be negative. Concerning the previous comment, when the positive values of Zim are used, the real part of complex permittivity would be positive as well.
- Figure 17 – omit the right Y-axis, since it is the same as the left one.
Author Response
Dear Alexander Code (MPhys) editor:
Many thanks for your email dated July 27, 2021. Based on the comments of the reviewers, the manuscript has been revised accordingly. We sincerely appreciate for reviewers’ careful reading and valuable comments to improve the quality of this paper. The revised parts are highlighted in the attached manuscript. In addition, we would like to provide the following responses to the reviewers’ comments.
Reviewer 2
Comments and Suggestions for Authors:
- The subject is suitable for the journal and is relevant to current interests.
Response: Thank you for your recognition and good comments. We sincerely appreciate for your careful reading and valuable comments. The comments are very helpful for revising and improving the paper. We have studied comments carefully and have made correction which we hope meet with approval. The main corrections in the paper can be seen from the revised manuscript. And the revised parts are highlighted in the attached manuscript.
- English should be improved.
Response: We sincerely appreciate for your careful reading and your good comments, and we have revised the full text according to your comments.
- Using word “electrochemical” is not appropriate within the whole paper. Electrochemistry refers to a relationship between electrical potential, as a measurable and quantitative phenomenon, and identifiable chemical change, with either electrical potential as an outcome of a particular chemical change, or vice versa. Since there is no chemical change associated with this investigation, the expression “electrical properties” must be used instead, even in the title.
Response: Thank you for your good comments and wise insights. The word “electrochemical” has been revised as “electrical properties” in the manuscript. Please see the title and the lines 13, 14, 18, 75, 90, 96, 200, 203, 216, 362, 393, 493, 498.
- The abbreviated labelling of the graphene oxide in the abstract is bit confusing. Once it is assigned as rGO, the second time only as GO. The same problem occurs in section 2.1.1. It is not clear whether GO refers to graphene oxide or graphite oxide. Unify the labelling or change the description!
Response: We sincerely appreciate for your careful reading and your good comments. We are sorry about the unclear abbreviated labelling. In this paper, the GO refers to graphene oxide and the rGO refers to the reduced graphene oxide. Moreover, the graphene oxide (GO) is obtained after the ultrasonication process for graphite oxide solution. The unclear abbreviated labelling is revised in the manuscript. Please see lines 11, 116, 117.
- Concerning table 2, it is strange that carbon content is much higher than the total solid content. Can you explain this discrepancy?
Response: We sincerely appreciate for your careful reading and your good comments. We are sorry about the unclear description in table 2. In this paper, the graphite oxide is black paste, which was consisted with solid graphite oxide and water. The solid content in table 2 represents the solid graphite oxide content in the graphite oxide paste. And the carbon content in table 2 represents the carbon content in solid graphite oxide. The unclear description is revised in table 2. Please see table 2, line 113.
- How was the specific surface area of GGBS determined? Please indicate the method (Blaine, BET).
Response: The GGBS used in this paper was purchased from Wuhan SinoCem Smartec Co., Ltd. And the specific surface area of GGBS was determined by Wuhan SinoCem Smartec Co., Ltd., confirming to the Chinese Standard GB/T 18046-2008. And the corresponding description has been added in this paper. Please see lines 102-104.
- What is the nature of graphene oxide? You are talking about powder, but table 2 indicates the appearance as black paste and solid content is only 43%. So, is it more like an aqueous suspension?
Response: Thank you for your careful reading, and we are sorry about the unclear description. The graphene oxide (GO) is obtained after the ultrasonication process for graphite oxide solution. And the detailed procedure is as follows: “Graphite oxide powder was magnetically stirred with deionized water for 30 min to get graphite oxide solution. Subsequently, the resulting solution (16.7 g/L) was ultrasonized 2 h with the power of 400 W and the frequency of 25 Hz. After ultrasonication, the well dispersed GO solution was obtained.” Please see lines 116-119. And the GO used in this paper was a uniformly dispersed suspension, which is similar to the GO used in the reference [29].
[29] Long, W. J.; Gu, Y. C.; Xing, F.; Khayat, K. H., Microstructure development and mechanism of hardened cement paste incorporating graphene oxide during carbonation. Cement & Concrete Composites 2018, 94, 72-84.
- The pH of graphite oxide is quite low. Was this taken into account in the composition of the AASC mixtures? rGO is neutralised during preparation by the addition of NaOH but the GO solution remained strongly acidic. In that case you have different mixture proportions of the AAS matrix between R0, G* and rG* samples.
Response: We sincerely appreciate for your careful reading. In this paper, we used NaOH and Na2SiO3 to activate the GGBS to obtain the AASC. Thus, the AASC will remain strongly alkaline Moreover, we guaranteed the same solid content of GO and rGO in AASC, and the mechanical performance and the corresponded microscopic experimental analysis were tested after hydration of 28 days. Therefore, the pH of GO and rGO will not influence the properties of AASC. And, we referred to two relevant literatures. Please see the references [9], [17] and [18].
[9] Yan, S.; He, P. G.; Jia, D. C.; Yang, Z. H.; Duan, X. M.; Wang, S. J.; Zhou, Y., Effect of reduced graphene oxide content on the microstructure and mechanical properties of graphene-geopolymer nanocomposites. Ceramics International 2016, 42, (1), 752-758.
[17] Zhu, X. H.; Kang, X. J.; Yang, K.; Yang, C. H., Effect of graphene oxide on the mechanical properties and the formation of layered double hydroxides (LDHs) in alkali-activated slag cement. Construction and Building Materials 2017, 132, 290-295.
[18] Long, W. J.; Ye, T. H.; Luo, Q. L.; Wang, Y.; Mei, L., Reinforcing Mechanism of Reduced Graphene Oxide on Flexural Strength of Geopolymers: A Synergetic Analysis of Hydration and Chemical Composition. Nanomaterials 2019, 9, (12).
- Data for the rG3 mixture are missing in Table 3 and Table 4.
Response: The rG3 mixture has been added in Table 3 and Table 4. Please see lines 148-149 and 258-259.
- What was the temperature and device geometry used during rheological tests? What is the wavelength of XRD source? Description of the methodology is not sufficient.
Response: Considering the reviewer’s suggestion, the temperature and device geometry used during rheological tests were added as follows: “In this study, the RM 100 touch device manufactured by Lamy Rheology Instruments (Lamy Rheology Instruments Company, Champagne-au-Mont-d’Or, France) was used to evaluate the rheological properties of the fresh pastes. The RM 100 touch device was shown in Figure 2. The temperature during the test was 25°C, and the measuring system used in this study was DIN-2.” Please see lines 152-156, 170-171. And as the reviewer suggested, the wavelength of XRD has been added in section 2.2.5 as follows: “The XRD experiment parameters are as follows: the test employed Cu Kα radiation (λ = 1.54 Å with a fixed divergence slit size of 0.5 °C and a rotating sample stage), and the range of 3°≤ 2θ ≤ 80°, scan rate of 0.02° per step, a voltage of 40 kV and current of 40 mA” Please see lines 185-188.
- Specify the title of 3.1. subsection!
Response: As the reviewer suggested, the “3.1. subsection” has been revised as “3.1. Characterization of GO and rGO” Please see line 221.
- The resolution of TEM images in Fig. 6 must be improved. The labels in the micrographs are not legible even at high magnification.
Response: Considering the reviewer’s suggestion, the resolution of TEM images has been improved. Please see lines 246-247.
- Row 251 – AASC cannot be regarded as Newtonian fluid because Bingham model describes non-Newtonian behaviour (it possesses a yield stress). However, if we take into account a measurement error, the yield stress of the samples R0 and G1 is so low that it can be considered a zero. Therefore, the Newtonian behaviour could be practically attributed only to these two mixtures.
Response: Thank you for this valuable comment. The rheological test in this paper was conformed to the Bingham model. And the incorrect description has been revised. Please see lines 253-255.
- Rows 262–264 – probably tossed meaning of the sentence because yield stress represents the maximum stress that prevent the plastic deformation and viscosity represents the resistance to flow and refers to the slope of the Bingham curve.
Response: Thank you for this valuable and insightful comment. The sentence “Generally, the plastic viscosity represents the maximum stress to prevent the plastic de-formation of the pastes, while the yield stress corresponds to the deformation speed of the pastes” has been revised as “Generally, yield stress represents the maximum stress that prevent the plastic deformation and viscosity represents the resistance to flow and refers to the slope of the Bingham curve.” Please see lines 265-267.
- Can author provide any reference to the statement concerning the rows 342–345? Moreover, this sentence makes no sense because it is grammatically incorrect.
Response: We sincerely appreciate for your wise insight. In this paper, we mainly discussed the reinforcing effect of rGO on AASC, and we have given the corresponding analysis in section 3.3 and section 3.4. And we agree with reviewer that this sentence makes no sense, and we have revised this paragraph. Please see lines 336-340.
- Rows 355–356 – Please, explain in more detail how the thermogravimetric measurement explains the reinforcing effect of rGO nanosheet compared to GO nanosheet. In my opinion, thermogravimetry provides no information about the morphology or the mechanical properties.
Response: Thank you for this valuable and insightful comment. I am sorry for my unclear description, and I also agree with you that thermogravimetry provides no information about the morphology or the mechanical properties.
In the section 3.5, TGA analysis plays as an aider to better understand the variation of mechanical properties, which is used to assist to explain the enhancement effect of rGO on AASC from the micro level. Generally, the compressive strength of AASC relates to the content of hydration products, and the TGA test can give the mass loss of the hydration products thereby helping to explain the enhancement effect of rGO and GO. This refers to the references [36-39]. And in the TGA analysis, the figure reveals that the main mass loss in the temperature range of 50-200 °C of the samples related to the C-A-S-H gels [40, 41], and the hydration products is related to the mechanical strength of the samples. It can be seen from the figures, when the nanomaterials dosage is 0.3% wt, the C-A-S-H gels peak of G3 is obviously lower than rG3. Therefore, it can be seen that when the nanomaterials dosage is 0.3% wt, the reinforcing effect of rGO nanosheet is more obvious than the GO nanosheet from the micro-product perspective analysis. And the analysis is corresponded to the results of compressive strength. In the paper, the unclear description has been revised. Please see lines 353-358.
[36] Wu, L.; Liu, L.; Gao, B.; Munoz-Carpena, R.; Zhang, M.; Chen, H.; Zhou, Z.; Wang, H., Aggregation kinetics of graphene oxides in aqueous solutions: experiments, mechanisms, and modeling. Langmuir 2013, 29, (49), 15174-81.
[37] Li, X. Y.; Korayem, A. H.; Li, C. Y.; Liu, Y. M.; He, H. S.; Sanjayan, J. G.; Duan, W. H., Incorporation of graphene oxide and silica fume into cement paste: A study of dispersion and compressive strength. Construction and Building Materials 2016, 123, 327-335.
[38] Mohammed, A.; Sanjayan, J. G.; Nazari, A.; Al-Saadi, N. T. K., The role of graphene oxide in limited long-term carbonation of cement-based matrix. Construction and Building Materials 2018, 168, 858-866.
[39] Lin, J. L.; Shamsaei, E.; de Souza, F. B.; Sagoe-Crentsil, K.; Duan, W. H., Dispersion of graphene oxide-silica nanohybrids in alkaline environment for improving ordinary Portland cement composites. Cement & Concrete Composites 2020, 106.
[40] Ye, H. L.; Huang, L.; Chen, Z. J., Influence of activator composition on the chloride binding capacity of alkali-activated slag. Cement & Concrete Composites 2019, 104.
[41] Ye, H. L.; Cai, R. J.; Tian, Z. S., Natural carbonation-induced phase and molecular evolution of alkali-activated slag: Effect of activator composition and curing temperature. Construction and Building Materials 2020, 248.
- Row 372 (and further on) – replace the word “polymerization” rather with “polycondensation”. Polymerization is a radical reaction which is associated with chaining of organic alkenes. Moreover, the molecular structure of C-A-S-H and C-S-H gels is not polymeric but rather oligomeric.
Response: Thank you for this valuable and insightful comment. We agree with reviewer that the word “polymerization” has been revised as “polycondensation” in manuscript. Please see lines 158, 271, 299, 313, 371, 379 and 381
- The Zimshould be positive. In my opinion, it refers to the capacitive reactance which has always a positive value. If it was negative, then it would refer to inductance and this is not this case. Change the sign!
Response: We sorry for this mistake. As the reviewer suggested, we have checked the data, the Zim is positive. The corresponded figures have been revised. Please see lines 405-406, 414-415, 416-417, 418-419.
- Equations 2, 3, and 4 are incorrect. The square root is missing in Eq. 2. The negative sign before Z (Eq. 3 and 4) must be omitted because dielectric constant cannot be negative. Concerning the previous comment, when the positive values of Zimare used, the real part of complex permittivity would be positive as well.
Response: Thank you for your wise insight. I am sorry for the editing error in the formula, I have revised these equations, and the data have been checked again. Please see Eq. 2, Eq. 3 and Eq. 4.
- Figure 17 – omit the right Y-axis, since it is the same as the left one.
Response: As the reviewer suggested, the right Y-axis of Fig. 17 has been omitted. Please see lines 487-489.
Reviewer 3 Report
The paper deals with the properties of alkali activated slag-based mixtures containing graphene oxide or reduced graphene oxide. The topic is very interesting and the paper is well written. At the same time, the research design is appropriate and the techniques used are worthy of interest.
Here my minor comments:
- in the abstract the acronym rGO is not define
- as Provis & van Deventer claim, "alkali activated materials are not geopolymers". Please, use a proper term to define your material. If slag (with high CaO amout) is used, we generally talk about alkali activated materials
- 3.1 subsection? Maybe it is a typo
- Table 1: please revise superscript and subscript. Moreover, express the specific surface without digits
- Introduction: I recommend also this recent paper about GO in alkali activated slag based mixtures:
- D'Alessandro et al. "Self-sensing properties of green Alkali-activated binders with carbon-based nanoinclusions", 2020, Sustainability, Volume 12, Issue 23, Pages 1 - 13
- Line 288-292: I think that these sentences, even if correct, don't increase the knowledge on this topic and are a bit off topic
Author Response
Dear Alexander Code (MPhys) editor:
Many thanks for your email dated July 27, 2021. Based on the comments of the reviewers, the manuscript has been revised accordingly. We sincerely appreciate for reviewers’ careful reading and valuable comments to improve the quality of this paper. The revised parts are highlighted in the attached manuscript. In addition, we would like to provide the following responses to the reviewers’ comments.
Reviewer 3
Comments and Suggestions for Authors:
The paper deals with the properties of alkali activated slag-based mixtures containing graphene oxide or reduced graphene oxide. The topic is very interesting and the paper is well written. At the same time, the research design is appropriate and the techniques used are worthy of interest.
Response: Thank you for your recognition and good comments. We sincerely appreciate for your careful reading and valuable comments. The comments are very helpful for revising and improving the paper. We have studied comments carefully and have made correction which we hope meet with approval. The main corrections in the paper can be seen from the revised manuscript. And the revised parts are highlighted in the attached manuscript.
Here my minor comments:
- in the abstract the acronym rGO is not define
Response: We sincerely appreciate for your careful reading and your good comments. We are sorry about the unclear abbreviated labelling in abstract. The unclear abbreviated labelling has been revised in abstract. Please see line 11.
- as Provis & van Deventer claim, "alkali activated materials are not geopolymers". Please, use a proper term to define your material. If slag (with high CaO amout) is used, we generally talk about alkali activated materials
Response: We sincerely appreciate for your wise insight. As the reviewer suggested, we have revised “geopolymers” as “alkali activated materials”. Please see lines 318-321, 325, 336-338, 433
- 1 subsection? Maybe it is a typo
Response: As the reviewer suggested, the “3.1. subsection” has been revised as “3.1. Characterization of GO and rGO” Please see line 221.
- Table 1: please revise superscript and subscript. Moreover, express the specific surface without digits
Response: We sincerely appreciate for your careful reading and your wise insight. As the reviewer suggested, the superscript and subscript in Table 1 have been revised. Please see lines 111-112. And the GGBS used in this paper was purchased from Wuhan SinoCem Smartec Co., Ltd. And the specific surface area of GGBS was determined by Wuhan SinoCem Smartec Co., Ltd., confirming to the Chinese Standard GB/T 18046-2008. And the corresponding description has been added in this paper. Please see lines 102-104.
- Introduction: I recommend also this recent paper about GO in alkali activated slag based mixtures: D'Alessandro et al. "Self-sensing properties of green Alkali-activated binders with carbon-based nanoinclusions", 2020, Sustainability, Volume 12, Issue 23, Pages 1 – 13
Response: Thank you for this valuable and insightful comment. We have added this recent paper in introduction. Please see the reference [10], line 49.
[10] D'Alessandro, A.; Coffetti, D.; Crotti, E.; Coppola, L.; Meoni, A.; Ubertini, F., Self-Sensing Properties of Green Alkali-Activated Binders with Carbon-Based Nanoinclusions. Sustainability-Basel 2020, 12, (23).
- Line 288-292: I think that these sentences, even if correct, don't increase the knowledge on this topic and are a bit off topic
Response: Thank you for this valuable and insightful comment. We agree with reviewer that these sentences don't increase the knowledge on this topic and are a bit off topic, and these sentences have been omitted. Please see lines 289-291.
Round 2
Reviewer 2 Report
The manuscript was revised according to the suggestions and comments andI am satisfied with it. However, I found two minor mistakes.
row 253 - Pearson's coefficient is square - please use superscript for R2
Capture of Fig 18 - please omit description of y-axes because there is only one now.
Author Response
Response:
Dear Alexander Code (MPhys) editor:
Many thanks for your email dated July 30, 2021. Based on the comments of the reviewers, the manuscript has been revised accordingly. We sincerely appreciate for reviewers’ careful reading and valuable comments to improve the quality of this paper. The revised parts of round 2 are highlighted in the attached manuscript. In addition, we would like to provide the following responses to the reviewers’ comments.
Reviewer 2 - Round 2
Comments and Suggestions for Authors:
The manuscript was revised according to the suggestions and comments and I am satisfied with it. However, I found two minor mistakes.
Response: Thank you for your recognition and good comments. We sincerely appreciate for your careful reading and valuable comments. And we are very grateful for your recognition of our first revision work. Your comments are very helpful to improve the quality of this paper. We have studied comments carefully and have made correction which we hope meet with approval. The main corrections in the paper can be seen from the revised manuscript – round 2, and the revised parts are highlighted in the attached manuscript.
- row 253 - Pearson's coefficient is square - please use superscript for R2.
Response: We sincerely appreciate for your careful reading. As the reviewer suggested, the “R2” has been revised as “R2” Please see line 253.
- Capture of Fig 18 - please omit description of y-axes because there is only one now.
Response: Thank you for this valuable and insightful comment. Considering the reviewer’s suggestion, the description of y-axes in Figure. 18 has been omitted, which shown as follows. “Figure 18. Electrical conductivity of the samples with GO and rGO: (a) 20 Hz; (b) 100 Hz; (c) 1000 Hz; (d) 100000 Hz.” Please see line 488-489.
Reviewer 3 Report
All the reviewers' suggestions have been addressed
Author Response
Response:
Dear Alexander Code (MPhys) editor:
Many thanks for your email dated July 30, 2021. Based on the comments of the reviewers, the manuscript has been revised accordingly. We sincerely appreciate for reviewers’ careful reading and valuable comments to improve the quality of this paper. The revised parts of round 2 are highlighted in the attached manuscript. In addition, we would like to provide the following responses to the reviewers’ comments.
Reviewer 3 – Round 2
All the reviewers' suggestions have been addressed
Response: Thank you for your recognition and good comments. We sincerely appreciate for your careful reading and valuable comments. The comments are very helpful for revising and improving the paper. We have studied comments carefully and have made correction which we hope meet with approval. The main corrections in the paper can be seen from the revised manuscript. And the revised parts are highlighted in the attached manuscript.